# Dy_4_, Dy_5_, and Ho_2_ Complexes of an *N_3_O_2_* Aminophenol Donor: A Dy_5_-µ_3_-Peroxide Single Molecule Magnet

**DOI:** 10.3390/ijms24109061

**Published:** 2023-05-21

**Authors:** Julio Corredoira-Vázquez, Paula Oreiro-Martínez, Daniel Nieto-Pastoriza, Ana M. García-Deibe, Jesús Sanmartín-Matalobos, Matilde Fondo

**Affiliations:** 1Departamento de Química Inorgánica, Facultade de Química, Universidade de Santiago de Compostela, 15782 Santiago de Compostela, Spain; 2Phantom-g, CICECO—Aveiro Institute of Materials, Department of Physics, University of Aveiro, 3810-193 Aveiro, Portugal; 3Institute of Materials (iMATUS), Universidade de Santiago de Compostela, 15782 Santiago de Compostela, Spain

**Keywords:** lanthanoid, SMM, peroxide ligand, *N*_3_*O*_2_ aminophenol

## Abstract

The reactivity of the new flexible potentially pentadentate *N_3_O_2_* aminophenol ligand H_4_L^r^ (2,2′-((pyridine-2,6-diylbis(methylene))bis(azanediyl))diphenol) towards different dysprosium salts and holmium(III) nitrate was investigated. Accordingly, this reactivity seems to greatly depend on the metal ion and salt employed. In this way, the reaction of H_4_L^r^ with dysprosium(III) chloride in air leads to the oxo-bridged tetranuclear complex [Dy_4_(H_2_L^r^)_3_(Cl)_4_(μ_3_-O)(EtOH)_2_(H_2_O)_2_]·2EtOH·H_2_O (**1**·2EtOH·H_2_O), while the same reaction just changing the chloride salt by the nitrate one renders the peroxo-bridged pentanuclear compound [Dy_5_(H_2_L^r^)_2_(H_2.5_L^r^)_2_(NO_3_)_4_(µ_3_-O_2_)_2_]·2H_2_O (**2**·2H_2_O), where both peroxo ligands seem to come from the fixation and reduction of atmospheric oxygen. However, if holmium(III) nitrate is used instead of dysprosium(III) nitrate, no evidence of a peroxide ligand is observed, and the dinuclear complex {[Ho_2_(H_2_L^r^)(H_3_L^r^)(NO_3_)_2_(H_2_O)_2_](NO_3_)} 2.5H_2_O (**3**·2.5H_2_O) is isolated. The three complexes were unequivocally characterized by X-ray diffraction techniques, and their magnetic properties were analyzed. Thus, while the Dy_4_ and Ho_2_ complexes do not show magnet-like behavior even in the presence of an external magnetic field, **2**·2H_2_O is a single molecule magnet, with an *U*_eff_ barrier of 61.2 K (43.2 cm^−1^). This is the first homonuclear lanthanoid peroxide SMM, which also shows the highest barrier among the reported 4f/3d peroxide zero field SMMs to date.

## 1. Introduction

Molecular magnetism is a growing field in which single-molecule magnets (SMMs) occupy a preferential place. The great interest in SMMs is based on their potential applications, which include ultrahigh-density information storage or quantum computing [1,2]. In contrast to classical magnets, the molecular nature of SMMs offers unique properties that could enable unprecedented information storage speed, and processing densities [2]. The main requirement to achieve this end is the ability to block the magnetization of the molecule at elevated temperatures, a goal that has so far not been achieved. The high intrinsic spin and anisotropy of lanthanoids make them preferred candidates in the search for molecule magnets with enriched properties. Accordingly, since the discovery of the first single ion magnet (SIM) in 2003 [3], the performance of SMMs has steadily improved. Thus, the best functioning magnets are mononuclear Dy^III^ compounds, so that, at present, the blocking temperature (*T_B_*) record of 80 K is held by [(Cp^iPr5^)Dy(Cp*)][B(C_6_F_5_)_4_] [4]. However, this kind of metallocene is unstable in air, and the *T*_B_ record for an air-stable compound is held by [Dy(bmbpen-F)Br] (bmbpen-F = N,N’-bis-(5-methyl-2-hydroxybenzyl)-N,N’-bis(5-fluoro-2-methylpyridyl)ethylenediamine) [5] at 36 K, although the highest energy barrier (>1800 K) for an air-stable SMM is owned by a Dy^III^ SIM with hexagonal bipyramidal geometry [6].

Therefore, as it is logical, and based on Long’s theory [7], it seems easier to control the anisotropy of mononuclear complexes, which leads to better magnetic results. However, even so, the performance of magnets with more than one metal center is also improving significantly. Thus, it has recently been described that the dinuclear mixed-valent complex [(Cp^iPr5^)_2_Dy_2_I_3_] [8], which features metal-to-metal bonding, shows an enormous coercive magnetic field, and hysteresis up to 80 K, although it is also unstable in air. In addition, the Dy–Sc nitride cluster fullerene Dy_2_ScN@C80-I_h_, with a nitrogen bridge between the Dy^III^ ions, presents an *U*_eff_ barrier of ca. 1735 K [9], very close to the record for an air-stable SIM. In addition, among air stable complexes, we have described the phenoxo-bridged dinuclear dysprosium complex with the highest energy barrier for spin reversal for this type of compound to date [10]. This was achieved with an *N*_4_*O*_3_ aminophenol donor, a ligand that also leads to fluoride-bridged complexes, some of them with unprecedented structural and magnetic features [11]. Accordingly, following this line of work, this paper describes the versatility in the coordination chemistry of a new pentadentate *N*_3_*O*_2_ aminophenol with Dy^III^ and Ho^III^, which includes the isolation of a rare Dy^III^ pentanuclear peroxo-complex, and the study of the magnetic properties of all the obtained compounds. 

## 2. Results and Discussion

### 2.1. Synthesis

H_4_L^r^, which is original from this work, is obtained by reducing the previously reported Schiff base H_2_L [12,13,14] with sodium borohydride, as summarized in Figure 1. The analytical and spectroscopic studies of H_4_L^r^ (Experimental section, and Appendix A) completely agreed with its isolation with high purity.

The reactivity of H_4_L^r^ towards lanthanoid salts in air seems to greatly depend on the lanthanoid employed but also on the anion of the metal salt, as shown in Figure 1. Accordingly, when the ligand reacts with dysprosium(III) chloride in a basic medium, single crystals of the Dy_4_ complex **1**·2EtOH·H_2_O with a µ_3_-oxide bridge are isolated. The composition of these crystals is approximate, given that SQUEEZE [15] has to be applied due to the great disorder of the trapped solvent. Thus, these crystals seem to lose the ethanol solvate on drying, and the elemental analyses of the dried crystals agree with formula **1**·5H_2_O.

When the aminophenol ligand reacts with dysprosium(III) nitrate instead of chloride, the peroxo-bridged Dy_5_ complex **2**·2H_2_O is obtained. Nevertheless, if holmium(III) nitrate is used in the place of the dysprosium(III) one, the dinuclear holmium compound **3**·2.5H_2_O, without any evidence of peroxide as ligand, is separated. Accordingly, this indicates that the formation of the peroxo donor is achieved in the presence of nitrate and Dy^III^ but not of Ho^III^. This may seem somewhat strange since one would expect the chemical behavior of Dy^III^ and Ho^III^ to be similar. In this respect, we should point out that a search was made in CSD [16] for crystallographically characterized peroxo-lanthanoid complexes in which there is no doubt as to the nature of the oxygenated ligand. This search show that homonuclear peroxo-complexes has been obtained from air with Ce^IV^ [17,18,19,20,21,22], Nd^III^ [23,24,25], Sm^III^ [26,27,28], Eu^III^ [23,24,26], Gd^III^ [26], Tb^III^ [29], Dy^III^, [30], Yb^III^ [23,31,32,33,34], and Lu^III^ [35] ions, in addition to 3d/4f M_3_Ln_3_ (M = Ni, Cu or Zn; Ln = Gd, Tb, or Dy) [36,37] and Zn_4_Ln_7_ (Ln = Gd, Dy) peroxo compounds [38]. However, as far as we know, no holmium peroxo-complexes have been obtained by interaction with oxygen. In addition, some other lanthanoid peroxo-complexes of the mentioned ions or La^III^ have been mostly isolated from peroxides as reagents [39,40,41,42]. Among these, it is worth noting that there is a series of peroxo-complexes of Gd, Tb, Dy, and Er with the same ligand, but not the Ho one [42]. Accordingly, based on this literature search, in which not a single peroxo-Ho complex was found, the results described here are not surprising, and, as expected, the interaction of Ho^III^ with the peroxide ligand does not appear to be favored.

Attempts were made to determine the origin of the peroxide species. Consequently, the reaction of H_4_L^r^ and Dy(NO_3_)_3_·6H_2_O was repeated in a strict inert atmosphere under Ar. Unfortunately, it was not possible to obtain single crystals of this sample of sufficient quality to be solved. However, the elemental analysis of this compound seems to be in agreement with the empirical formula {[Dy_2_(H_2_L^r^)(H_3_L^r^)(NO_3_)_2_(H_2_O)_2_](NO_3_)}·2EtOH·H_2_O (**4·**2EtOH·H_2_O), similar to that of the holmium complex **3**·2.5H_2_O. Obviously, in the absence of a single crystal structure, it does not make sense to speculate more on the possible structure of this compound, but IR spectroscopy seems to help determine the nature of the species. Thus, the IR spectrum of **2**·2H_2_O (Appendix A) shows a band at 831 cm^−1^, absent in the spectra of the nitrate holmium complex **3**·2.5H_2_O and of the nitrate dysprosium compound **4·**2EtOH·H_2_O. The position, shape, and intensity of this band seems to agree with the vibration of the peroxide ligand [25,37,38]. The presence of this donor is also confirmed by the Raman spectrum, which shows a weak peak at 831 cm^−1^ (Appendix A), whose shape and intensity are similar to that previously described for other lanthanoid peroxo-complexes [18,37]. In addition, the IR spectroscopy seems to indicate that the peroxo group does not form in the absence of air, as this band is not present in the spectrum of **4·**2EtOH·H_2_O (Appendix A). However, additionally, when the mother liquor from the synthesis of **4·**2EtOH·H_2_O in inert atmosphere is left in air, a small portion of a new solid **5** precipitates, whose IR spectrum is identical to that of **2**·2H_2_O between 2000 and 500 cm^−1^, and it again shows the presence of the band at 831 cm^−1^ (Appendix A). Accordingly, the peroxide ligand seems to come from air reduction. 

### 2.2. X-ray Diffraction Studies 

Single crystals of **1**·2EtOH·H_2_O-**3**·2.5H_2_O were obtained as detailed above. The experimental details of data acquisition and resolution are summarized in Appendix A.

Dy_4_(H_2_L^r^)_3_(Cl)_4_(μ_3_-O)(EtOH)_2_(H_2_O)_2_]·2EtOH·H_2_O (**1**·2EtOH·H_2_O). An ellipsoid diagram for **1** is shown in Figure 1, and their main bond distances and angles are listed in Appendix A. 

The unit cell of **1**·2EtOH·H_2_O contains neutral molecules of [Dy_4_(H_2_L^r^)_3_(Cl)_4_(μ_3_-O)(EtOH)_2_(H_2_O)_2_] (**1**) and water and ethanol as solvates. It should be noted that SQUEEZE [15] was applied to these crystals to correctly solve the structure due to the high disorder of the solvent. Accordingly, the composition of the solvates in the crystals is inexact. 

[Dy_4_(H_2_L^r^)_3_(Cl)_4_(μ_3_-O)(EtOH)_2_(H_2_O)_2_] does not contain symmetry elements. However, in order to simplify its description, the tetranuclear compound can be considered as formed from 3 mononuclear building blocks, each of which uses some of its donor atoms to bind a fourth Dy^III^ ion. Two of these three mononuclear units, those containing Dy1 and Dy3, which we will therefore call **1**.1 and **1**.3, are chemically equivalent but crystallographically different, and written as the neutral fragments [Dy(H_2_L^r^)(Cl)(H_2_O)] (Figure 2a). The mononuclear unit containing Dy2, which we call **1**.2, is the monoanionic fragment [Dy(H_2_L^r^)(Cl)_2_(EtOH)]^−^ (Figure 2b).

In all the mononuclear moieties, the phenol oxygen atoms of the ligand are deprotonated, and the dianionic aminophenol acts as a pentadentate *N*_3_*O*_2_ donor. In addition, in fragments **1**.1 and **1**.3, a water molecule and a chloride ligand also bind the metal ion, leading to coordination number 7. For fragment **1**.3, two chloride anions and an ethanol donor from the solvent of the reaction are also coordinated to the dysprosium atom, leading to an octacoordinated environment. 

These three mononuclear fragments are joined among them and to Dy4 by means of various bridges, completing their coordination spheres. Accordingly, Dy1 reachs coordination number 8 by binding to an O^2−^ bridge (O1), while Dy3 reachs coordination number 9 by also binding to O1 and to a phenolate oxygen atom (O11) of one of the ligand arms of unit **1**.1. Therefore, between units **1**.1 and **1**.3, there is a double oxygen bridge (phenolate O11 and oxo O1, Figure 3), and this *Dy*_2_*O*_2_ core shows Dy-O-Dy angles close to 110°. 

Units **1**.1, **1**.2, and **1**.3 employ all their phenolic oxygen atoms that were not previously used in coordinated bonds (two in **1**.2 and **1**.3, and one in **1**.1, a total of five) to bind the fourth dysprosium ion Dy4 (Figure 1 and Figure 3). This Dy4 center is further bonded to the oxo bridge O1 and to one ethanol ligand, thus reaching coordination number 7. Accordingly, as a consequence of the described features, as shown in Figure 3, the pairs Dy1 and Dy4, and Dy2 and Dy4, are also double bridged, and the *Dy*_2_*O*_2_ groups show distances comparable to those found between Dy1 and Dy3, with similar Dy-O-Dy angles. However, Dy3 and Dy4 are triple bridged by two phenolic oxygen atoms and by an oxygen atom of the oxo group (O1), which at this point acts as a μ_3_ bridge, linking Dy1, Dy3, and Dy4. This triple bridged *Dy*_2_*O*_3_ core, as would be expected, shows the shortest distance between dysprosium atoms (Figure 3), with much shorter Dy-O-Dy angles, between 96.3 and 98.3°. 

As a result of the situation described, the three [H_2_L^r^]^2−^ ligands show two different coordination modes. In all three cases, it acts as pentadentate, but in one case (when the ligand inserts Dy1 in its *N*_3_*O*_2_ pocket) it is a bridging μ_2_ donor, and in two others (when it inserts Dy2 or Dy3 in the pocket), a μ_3_ donor (Figure 4).

According to the discussed features, the coordination numbers of dysprosium atoms are different: 8 for Dy1 and Dy2, 9 for Dy3, and 7 for Dy4. Calculations of the distortion of the polyhedra with respect to the ideals for 8, 9, or 7 vertices with the SHAPE program [43] (Appendix A) show that the geometry that best represented the environment of Dy1 and Dy2 is a triangular dodecahedron, for Dy3 a “muffin”, and for Dy4 a capped octahedron. All distances and bond angles in these polyhedra are within their normal range [10,13,14] and do not merit further consideration.

Finally, it should be noted that one of the amine nitrogen atoms and one of the chlorine ligands per tetranuclear molecule are involved in a weak intermolecular hydrogen bond, which expands this tetranuclear unit into a zig-zag chain. 

[Dy_5_(H_2_L)_2_(H_2.5_L)_2_(NO_3_)_4_(µ_3_-O_2_)_2_]·2H_2_O (**2**·2H_2_O). An ellipsoid diagram for **2** is shown in Figure 5, and their main bond distances and angles are listed in Appendix A.

The unit cell of **2**·2H_2_O contains molecules of the neutral pentanuclear complex [Dy_5_(H_2_L^r^)_2_(H_2.5_L^r^)_2_(NO_3_)_4_(µ_3_-O_2_)_2_], together with water as a solvate. The asymmetric unit of the crystal contains only half of the complex molecule, the other half being generated by an improper rotation axis (−x + 1, y, −z + ½) passing through Dy3. Thus, each molecule of this neutral complex can be understood as two dinuclear [Dy(H_2_L^r^)(NO_3_)(µ_2_-O_2_)Dy(H_2.5_L^r^)(NO_3_)]^1.5−^ blocks (Figure 6), which join a fifth Dy^3+^ ion (Dy3) between them, using four phenolate groups (one per aminophenol ligand, O11 and O21), and the two (one per block) peroxide ligands (O1-O2), as shown in Figure 7. This leads to the aminophenol acting as a µ_2_ bridge but in a different way (µ_2_-κ^5^:κ^1^, Figure 7) from **1**.

In turn, the respective dinuclear blocks can be considered constructed from two mononuclear units, [Dy(H_2.5_L^r^)(NO_3_)]^0.5−^ and [Dy(H_2_L^r^)(NO_3_)]^−^, which contain Dy1 and Dy2, respectively. The main difference between the units lies in the fact that one of the phenolic oxygen atoms (O12) is semi-protonated in one of the units but not in the other. At the same time, in each mononuclear unit, the aminophenol ligand links a Dy^III^ center in its *N*_3_*O*_2_ pocket, as in **1**. Each metal center also binds a nitrate ligand, which acts as a bidentate chelate donor (Figure 6).

In addition, both oxygen atoms of the peroxide group (O1 and O2) are coordinated in side-on mode to the two dysprosium atoms, with a partial coordination mode µ_2_-κ^2^:κ^2^ in each dinuclear block (Figure 6). Thus, in these dinuclear entities, the dysprosium atoms are nonacoordinated, in *N*_3_*O*_6_ environments. Calculations with the SHAPE program (Appendix A) [43] indicate that the geometry around the dysprosium atoms is spherical capped square antiprism.

The central dysprosium atom (Dy3, see Figure 7) is in a highly distorted *O*_8_ environment. This is achieved by means of the four mentioned bridging phenolate O-atoms, and two peroxide ligands acting as µ_3_-κ^2^:κ^2^:κ^2^ bridges, and according to SHAPE measurements (Appendix A) [43], this environment can be described as a biaugmented trigonal prism.

The coordination mode of the O_2_^2−^ ligand is quite common for lanthanoid complexes with peroxide as donor [25,36,37,38,42], and all the distances between this donor and the pentadentate ligand are within their normal ranges [10,13,14,36,37,38,42]. In addition, the dysprosium ions within the dinuclear subunits bridged by the O_2_^2−^ donor are at distances Dy1···Dy2 of 4.2636(6) Å, with Dy1-O_peroxide_-Dy2 angles of ca. 129°. A double hydrogen bond between one oxygen atom of the nitrate group joined to Dy1 and one amine nitrogen atom joined to Dy2 and vice versa (Figure 6) also contributes to shortening this distance. Dy1···Dy3 and Dy2···Dy3 are triple bridged by the peroxide and one phenol oxygen atom (Figure 7). This leads to shorter Dy1···Dy3 and Dy2···Dy3 distances, ca. 3.6 Å, with Dy-O-Dy angles in the range 97.6–103.1°. 

[Ho_2_(H_2_L^r^)(H_3_L^r^)(NO_3_)_2_(H_2_O)_2_](NO_3_)·2·5H_2_O (**3**·2.5H_2_O). The unit cell of **3**·2.5H_2_O contains [Ho_2_(H_2_L^r^)(H_3_L^r^)(NO_3_)_2_(H_2_O)_2_]^+^ cations, NO_3_^-^ anions, and water as solvate. An ellipsoid diagram for **3** is shown in Figure 8, and the principal bond distances and angles are listed in Appendix A.

The dinuclear cationic complex [Ho_2_(H_2_L^r^)(H_3_L^r^)(NO_3_)_2_(H_2_O)_2_]^+^ can be considered assembled from two mononuclear blocks: cationic [Ho(H_3_L^r^)(NO_3_)(H_2_O)]^+^, which we call **3**.1, and neutral [Ho(H_2_L^r^)(NO_3_)(H_2_O)], which we call **3**.2. Both blocks are similar (Figure 9), the main difference being that in **3**.1 one of the phenolic oxygen atoms is protonated, while in **3**.2 both are deprotonated. Apart from this difference, in **3**.1 and **3**.2, the pentadentate ligand acts as usual, using its *N*_3_*O*_2_ donor set to link the Dy^III^ center.

In the two blocks, the holmium center additionally binds to one nitrate ligand in a bidentate chelate mode and to one water molecule (Figure 9). Units **3**.1 and **3**.2 join each other using each one of the blocks a deprotonated phenolic oxygen to act as a bridge between the Ho1 and Ho2 centers. Thus, the aminophenol ligands also acts as µ_2_-κ^5^:κ^1^ bridges, as in **2**. Therefore, this double phenolate bridge leads to a *Ho*_2_*O*_2_ core, with a Ho···Ho distance of 3.8091(2) Å, and Ho-O-Ho angles close to 109.7° (Figure 8).

As a result of the described situation, the metal ions are in both cases nonacoordinate, and calculations performed with the SHAPE program to see the degree of distortion with respect to an ideal 9-vertex polyhedron show that the geometry in the environment of the metal centers is somewhat different. Thus, for Ho1, the environment is close to a square capped antiprism, distorted toward a spherical tricapped trigonal prism, while for Ho2, the geometry is “muffin” distorted toward a square capped antiprism. The distances and bond angles of these olyhedral are within their normal range and do not merit further consideration [10].

Moreover, powder X-ray diffraction studies for crude samples of **2**·2H_2_O and **3**·2.5H_2_O (Appendix A) prove that these microcrystalline samples are the same compounds as the single crystals. The powder diffractogram for **1**·5H_2_O was not recorded, given that SQUEZZE was applied to the single crystals, and thus the solvates of the crystals for creating a theoretical diffractogram and those of the crude sample are different, thus preventing an accurate comparison of the experimental and theoretical diffractograms.

### 2.3. Magnetic Properties 

Susceptibility magnetic measurements were recorded for **1**·5H_2_O-**3**·2.5H_2_O between 2 and 300 K. The χ_M_*T* vs. *T* plots for the three complexes are shown in Figure 10.

The χ_M_*T* values at 300 K are 55.2 cm^3^Kmol^−1^ for **1**·5H_2_O, 66.5 cm^3^Kmol^−1^ for **2**·2H_2_O, and 28.8 cm^3^Kmol^−1^ for **3**·2.5H_2_O, which are reasonably close to the expected ones for two Ho^3+^ (28.14 cm^3^Kmol^−1^), four (56.68 cm^3^Kmol^−1^), or five (70.85 cm^3^Kmol^−1^) Dy^3+^ uncoupled ions at room temperature. For **1**·5H_2_O and **2**·2H_2_O, this curve continuously decreases with decreasing temperature, and the diminishment of the χ_M_*T* product is more pronounced below 50 K. For holmium complex **3**·2.5H_2_O, the χ_M_*T* is nearly constant from 300 to 100 K and then it decreases until 2 K, the diminishment being also more pronounced below 50 K. Accordingly, the continuous drop of the curves for **1**·5H_2_O and **2**·2H_2_O and the strong diminishment of the χ_M_*T* for **3**·2.5H_2_O at low temperature are attributed in all cases to thermal depopulation of the excited *M*_J_ levels, which leads to the existence of considerable single ion anisotropy.

The dependence of the magnetization with the magnetic field at 2 K (Figure 10, insets) shows that the reduced magnetization at 7 T tends to 19.5 Nµ_B_ for **1**·5H_2_O, 25.8 for **2**·2H_2_O, and 11.6 Nµ_B_ for **3**·2.5H_2_O. These values are quite similar to the expected ones for the average magnetization of a powder sample containing four or five highly anisotropic Dy^III^ ions (20 or 25 Nµ_B,_ respectively), and deviates slightly more from the expected value for the derivative containing two Ho^III^ (10 Nµ_B_) ions [44].

The dynamic magnetic properties for **1**·5H_2_O, **2**·2H_2_O, and **3**·2.5H_2_O were also studied. In a zero dc field, the out-of-phase signals of the ac susceptibility for **2**·2H_2_O show temperature and frequency dependence (Figure 11), with well-defined peaks for χ″_M_. Thus, **2**·2H_2_O is a peroxo-lanthanide single molecule magnet. Nevertheless, for **1·**5H_2_O and **2**·2H_2_O, no χ″_M_ peaks are seen without the application of a magnetic dc field. 

The fit of the Cole−Cole plot for **2**·2H_2_O with the generalized Debye model yields α parameters between 0.36 and 0.21, indicating a relatively wide distribution of relaxation times (Appendix A).

The relaxation time and the energy barrier for **2**·2H_2_O were extracted from the dependence of τ with the temperature plot (Figure 12). In this way, it should be noted that the dependence of χ″_M_ with *T* at different frequencies (Appendix A) agrees with fast relaxation of the magnetization via quantum channel (QTM). Therefore, attempts were made to fit the ln(τ) vs. 1/*T* plot considering all the spin-phonon mechanisms, in addition to QTM relaxation. However, the direct process can already be discarded, given that the ac data were recorded at a zero dc field. 

Thus, the best fit considering the other three processes, individually or grouped, is achieved with Orbach and QTM relaxation (Equation (1)).
(1)$$\tau {}^{-1}=\tau {{}_{0}}^{-1}e^{-U_{eff}/k_{B}T}+\tau {{}_{QTM}}^{-1}$$

The introduction of the Raman term, which should govern the relaxation at low temperature, does not improve the fitting and causes overparameterization. Therefore, the best fit renders the parameters *U*_eff_ = 61.2 K (43.2 cm^−1^), τ_0_ = 4.4 × 10^−7^ and τ_QTM_ = 4.2 × 10^−5^ s. Thus, a significant quantum channel is observed in this SMM. 

Attempts were made to compare these data with those of related lanthanoid peroxide SMMs, and the results are shown in Table 1. Accordingly, a search from CSD data [23] shows that there are not many crystallographically characterized lanthanoid peroxo-complexes, and among them, there are only seven whose behavior as a molecule magnet has been fully studied (Table 1) [36,37,38,42]. Additionally, there is another trinuclear neodymium compound that the authors say is a SMM, but they do not report any barrier, and actually the studies are done in a 1000 Oe field [25]. In addition, a Dy_6_Na_2_ metal complex with slow relaxation of the magnetization but without reported energy barrier has also been published [32].

For the seven complexes mentioned (Table 1), the coordination mode of the peroxide ligand is µ_3_-κ^2^:κ^2^:κ^2^, as in **2**·2H_2_O. Among these compounds, six are heteronuclear, and those Zn_x_Ln_y_ complexes can be really considered as diluted samples, given that the presence of zinc in heteronuclear Zn-Ln coordination compounds is well known to have a similar effect to magnetic dilution [45,46,47]. Thus, the comparison of the data in Table 1 clearly shows that **2**·2H_2_O is the first homonuclear 4f-peroxide SMM, given that the three other peroxide compounds with 4f elements being SMMs are mixed 3d/4f systems ([Zn_4_Dy_7_(L^1^)_8_(O_2_)_2_(OH)_4_(Cl)_4_(H_2_O)_4_]Cl, [Cu_3_Tb_3_(L^2^)_3_(O_2_)(PyCO_2_)_3_](OH)_2_(ClO_4_)_2_ and [Dy_3_Ni_3_(H_2_O)_3_(mpko)_9_(O_2_)(NO_3_)_3_](ClO_4_)). Besides, **2**·2H_2_O is the SMM with the highest zero-field barrier for all compounds, and this barrier is even higher than those for non-diluted samples with slow relaxation of magnetization and SMM-like behavior at different external fields. 

As previously discussed, the χ″_M_ vs. *T* plot for **2**·2H_2_O (Appendix A) indicates the existence of QTM, and this quantum channel can also be the reason for the non-observation of SMM behavior in **1**·5H_2_O and **3**·2.5H_2_O. Therefore, efforts have been made to eliminate this quantum channel. Thus, new *ac* measurements were recorded under different external dc fields at 3 K for **1**·5H_2_O and **3**·2.5H_2_O and at 6 K for **2**·2H_2_O. However, the application of different magnetic fields at 3 K did not lead to peaks in the χ″_M_ vs. frequency curves (Appendix A). Despite this, the field that leads to the slowest relaxation time was calculated (Appendix A), which is 2000 Oe. Accordingly, new measures were performed for **1**·5H_2_O and **3**·2.5H_2_O under this magnetic field at 8000 and 10,000 Hz as a function of the temperature, but no peaks were observed (Appendix A). For **2**·2H_2_O, the application of an external dc field does not improve the magnetic results (Appendix A), given that the highest relaxation time is obtained in the absence of any field. Therefore, it seems that an external field is unable to destroy or reduce the QTM effect for any of the complexes. 

## 3. Materials and Methods

The chemical reagents used in this work were purchased from commercial sources and used without further purification. Elemental analyses of C, H, and N were performed on a Thermo Scientific (Waltham, MA, USA) Flash Smart analyzer. Infrared spectra were registered in ATR mode on a Varian 670 FT/IR spectrophotometer in the range of 4000–500 cm^−1^. The ^1^H NMR spectrum of H_4_L^r^ was recorded on a Varian Inova 400 spectrometer.

### 3.1. Synthesis

H_4_L^r^ was prepared by reducing the previously reported Schiff base H_2_L [12,13,14], which was obtained as detailed before [12,13], and satisfactorily characterized.

H_4_L^r^: To a solution of H_2_L (1 g, 3.15 mmol) in methanol (120 mL), NaBH_4_ (0.25 g, 6.32 mmol) was added in small portions. The mixture was stirred for 30 min. Then, 30 mL of a solution of 10% H_3_PO_4_ in water was added to the initial mixture, and the pH was adjusted to 7.5 with a 10% NaOH water solution (20 mL). The obtained product was extracted with ethyl acetate (3 × 100 mL), and the organic phase was dried with sodium sulfate. This was removed, and the solution was concentrated in a rotary evaporator and dried with sodium sulfate. The dried solution was concentrated, and the precipitated solid was filtered and dried in air. Yield: 0.74 g (73%). MW = 321.35 g.mol^−1^. Elemental analysis (%): experimental: C 70.22%, N 12.86%, H 5.99%; calcd. for C_19_H_19_O_2_N_3_: C 70.95%, N 13.06%, H 5.91%. IR spectrum (ATR, $\tilde{\nu }$/cm^−1^): 1598 (νCN_Py_), 3401, 3435 (νNH), 3517 (νOH). ^1^H NMR (DMSO-*d*_6_, 400 MHz, δ in ppm): 9.30 (s, 2H, OH); 7.65 (t, 1H, H1); 7.19 (d, 2H, H2); 6.69 (d, 2H, H9); 6.59–6.53 (m, 2H), 6.43–6.38 (m, 4H) (2H6 + 2H7 + 2H8); 5.53 (s, 2H, NH); 4.39 (s, 4H, H4) (See Appendix A for numbering Scheme).

[Dy_4_(H_2_L^r^)_3_(Cl)_4_(μ_3_-O)(EtOH)_2_(H_2_O)_2_]·5H_2_O (**1**·5H_2_O): H_4_L^r^ (0.139 g, 0.433 mmol) and triethylamine (0.087 g, 0.865 mmol) were dissolved in absolute ethanol (35 mL). Dysprosium(III) chloride hexahydrate (0.161 g, 0.433 mmol) in absolute ethanol (10 mL) was added to the previous solution, and the mixture was stirred for 3 h. The resultant solution was left to slowly evaporate, and after two days, single crystals of [Dy_4_(H_2_L^r^)_3_(Cl)_4_(μ_3_-O)(EtOH)_2_(H_2_O)_2_]·2EtOH·H_2_O (**1**·2EtOH·H_2_O), suitable for single X-ray diffraction studies, were obtained. It should be noted that the solvate composition of these crystals was approximate since due to the large amount of disordered solvent present in the lattice, it was necessary to use SQUEZZE [15]. The crystals were filtered and dried in an oven, losing the ethanol solvate, and the characterization of the dried crystals seems to be in agreement with formula **1**·5H_2_O. Yield: 0.069 g (34%). MW: 1984.10 gmol^−1^. Elemental analysis (%): experimental: C 36.35, N 6.55, H 3.92; calcd for C_61_H_77_Cl_4_Dy_4_N_9_O_15_O: C 36.92, N 6.35, H 3.91. IR spectrum (ATR, $\tilde{\nu }$/cm^−1^): 1583 (νCN_Py_), 3200 (νNH, νOH).

[Dy_5_(H_2_L^r^)_2_(H_2.5_L^r^)_2_(NO_3_)_4_(µ_3_-O_2_)_2_] (**2**·2H_2_O): H_4_L^r^ (0.100 g, 0.311 mmol) and triethylamine (0.063 g, 0.622 mmol) in absolute ethanol (40 mL) were mixed with a solution of dysprosium(III) nitrate hexahydrate (0.142 g, 0.311 mmol) in absolute ethanol (25 mL). The mixture was stirred at room temperature for 3 h, and the resultant solution was filtered and left to slowly evaporate. After a few days, single crystals of [Dy_5_(H_2_L^r^)_2_(H_2.5_L^r^)_2_(NO_3_)_4_(µ_3_-O_2_)_2_]·2H_2_O, suitable for X-ray diffraction studies, were obtained. Yield (base on the metal): 0.032 g (21%). MW: 2435.9 gmol^−1^. Elemental analysis: experimental: C 37.37%, N 9.24%, H 2.99%; calcd for C_76_H_70_Dy_5_N_16_O_26_: C 37.47%, N 9.20%, H 2.99%. IR spectrum (ATR, $\tilde{\nu }$/cm^−1^): 831 (νO_2_^2−^), 1297, 1482 (νNO_3_), 1588 (νCN_Py_), 3257 (νNH), 3598 (νOH).

The same complex was obtained when Dy(NO_3_)_3_·6H_2_O and H_4_L^r^ were mixed in a 1.25:1 molar ratio. In this case, the yield increased to 28%.

[Ho_2_(H_2_L^r^)(H_3_L^r^)(NO_3_)_2_(H_2_O)_2_](NO_3_)·2.5H_2_O (**3**·2.5H_2_O): H_4_L^r^ (0.100 g, 0.311 mmol) and triethylamine (0.063 g, 0.622 mmol) in absolute ethanol (35 mL) were mixed with an absolute ethanol solution (10 mL) of holmium(III) nitrate pentahydrate (0.137 g, 0.311 mmol). The mixture was stirred for 3 h, and then the solution was concentrated in a rotary-evaporator to 1/3 of its initial volume. This solution was left to slowly evaporate, and after a few days, single crystals suitable for X-ray diffraction studies of [Ho_2_(H_2_L^r^)(H_3_L^r^)(NO_3_)_2_(H_2_O)_2_](NO_3_)·2.5H_2_O were isolated. The crystals were filtered and dried in an oven. Yield: 0.108 g (27%). MW = 1236.64 gmol^−1^. Elemental analysis (%): experimental: C 36.15, N 9.82, H 3.53; calcd. for C_38_H_44_Ho_2_N_9_O_17.5_: C 36.87, N 10.19, H 3.56. IR spectrum (ATR, $\tilde{\nu }$/cm^−1^): 1299, 1480 (νNO_3_), 1599 (νCN_Py_), 3239, 3268 (νNH), 3346 (νOH).

[Dy_2_(H_2_L^r^)(H_3_L^r^)(NO_3_)_2_(H_2_O)_2_](NO_3_)·2EtOH·H_2_O (**4**·2EtOH·H_2_O): The same reaction to obtain **2**·2H_2_O was repeated in an inert Ar atmosphere. A deep red solid precipitate was filtered using Schlenk techniques and subsequently dried in air. Yield: 0.124 g (59%). MW = 1296.9 g.mol^−1^. Experimental: C 38.68, N 9.70, H 3.89; calcd. for C_42_H_53_Dy_2_N_9_O_18_: C 38.89, N 9.72, H 4.12. IR spectrum (ATR, $\tilde{\nu }$/cm^−1^): 1298, 1485 (νNO_3_), 1590 (νCN_Py_), 3254 (νNH), 3533 (νOH).

No single crystals of **4**·2EtOH·H_2_O were obtained from the mother liquor in an inert atmosphere or by recrystallisation of the solid from different dried solvents (MeOH, THF) under Ar.

When the mother liquor was open to air, a very small new fraction of a red solid **5** (2 mg) was isolated. IR spectrum (ATR, $\tilde{\nu }$/cm^−1^): 831 (νO_2_^2−^), 1296, 1482 (νNO_3_), 1588 (νCN_Py_), 3260 (νNH), 3595 (νOH).

### 3.2. Crystallographic Refinement and Structure Solution

The crystal data and some details of the refinement are summarized in Appendix A. Single crystals of **1**·2EtOH·H_2_O, **2**·2H_2_O and **3**·2.5H_2_O were obtained as previously detailed. Data were collected at 100 K on a Bruker D8 VENTURE PHOTON III-14 diffractometer, employing graphite monochromatized Mo-kα (λ = 0.71073 Å) radiation. SADABS [48] was used to apply multi-scan absorption corrections. These structures were solved by standard direct methods, employing SHELXT [49], and refined by full matrix least-squares techniques on *F*^2^, by means of SHELXL, from the program package SHELX-2018 [49]. The electron densities corresponding to the disordered guest molecules of **1**·2EtOH·H_2_O were flattened using the SQUEEZE option [15] of PLATON.

All non-hydrogen atoms corresponding to the complexes were refined anisotropically, but in some cases, as disordered atoms or solvates with low occupation sites, were isotropically treated. Hydrogen atoms bonded to C atoms were introduced in the structure factor calculations in geometrically idealized positions. Hydrogen atoms linked to oxygen and/or nitrogen atoms, with a partial occupation of 1, were mainly located in the corresponding Fourier maps, with the object of revealing the hydrogen bonds. In these cases, either they were freely refined or with thermal parameters derived from their parent atoms. When the hydrogen atoms could not be located in the Fourier map, they were fixed at rational positions.

The structures of **1**·2EtOH·H_2_O, **2**·2H_2_O and **3**·2.5H_2_O were deposited to the Cambridge Crystallographic Data Centre (CCDC) (Cambridge, UK) as a supplementary publication, No. 2240183–2240185.

### 3.3. Powder X-ray Diffraction Studies

The powder diffractograms for **2**·2H_2_O and **3**·2.5H_2_O were registered on a Philips diffractometer, with a control unity type “PW1710”, a vertical goniometer type “PW1820/00” and a generator type “Enraf Nonius FR590”, operating at 40 kV and 30 mA, using monochromated Cu-Kα (λ = 1.5418 Å) radiation. A scan in the range 2 < 2θ < 50° was performed, with t = 3 s and Δ*2θ* = 0.02°. LeBail refinement was obtained using HighScore Plus Version 3.0d.

### 3.4. Magnetic Measurements

Magnetic susceptibility dc and ac data for microcrystalline samples of the three complexes were collected using a PPMS Quantum Design susceptometer. The dc magnetic susceptibility data were registered under a magnetic field of 1000 Oe between 2 and 300 K. Magnetization measurements at 2.0 K were registered under magnetic fields ranging from 0 to 70,000 Oe. Diamagnetic corrections were estimated from Pascal’s Tables. Alternating current (ac) susceptibility measurements at zero dc field were recorded with an oscillating ac field of 3.5 Oe, and ac frequencies of 10,000 and 8000 Hz for **1**·5H_2_O and **3**·2.5H_2_O, and ac frequencies ranging from 50 to 10,000 Hz for **2**·2H_2_O in the 2–25 K temperature range. ac measurements at 3 K or 6 K were also recorded under magnetic fields ranging from 250 to 3000 Oe, and ac frequencies between 50 and 10,000 Hz for **1**·5H_2_O and **3**·2.5H_2_O, and between 50 and 1500 Hz for **2**·2H_2_O. New ac data were acquired for **1**·5H_2_O and **3**·2.5H_2_O under a dc field of 2000 Oe, at 8000 and 10,000 Hz in the 2–25 K temperature range.

## 4. Conclusions

The coordination chemistry of the new potentially pentadentate *N*_3_*O*_2_ aminophenol H_4_L^r^ with Dy^III^ and Ho^III^ was investigated. The crystal structures of the complexes reported herein show a high versatility of this ligand when it coordinates to Ln^III^ ions, with three different coordination modes. In addition, this chemistry greatly depends on the metal ion and salt employed. Accordingly, the oxo-bridged chloride Dy_4_ complex **1**·5H_2_O is isolated when hydrated dysprosium(III) chloride is used as the starting salt, but the Dy_5_ peroxide compound **2**·2H_2_O is obtained when the employed salt is dysprosium(III) nitrate. This peroxide ligand is not formed if Dy^III^ is changed by Ho^III^, and this later produces the Ho_2_ complex **3**·2.5H_2_O. The experiments carried out seem to indicate that the peroxide ligand in **2**·2H_2_O comes from air reduction. This latter complex is the first homonuclear peroxide lanthanoid SMM, with an energy barrier of 61.2 K. This barrier is the highest one reported to date among the scarcely reported zero-field peroxide SMMs containing a 4f metal, and among the few examples of peroxide Ln-containing complexes that present SMM-like behavior under a magnetic field, without a dilution-like magnetic effect. Accordingly, this work contributes not only to the knowledge of the coordination chemistry of lanthanoids with flexible pentadentate aminophenol donors but also to increasing the limited number of compounds containing a 4f ion with a peroxide ligand that behave as magnets, either at zero field or in the presence of an external magnetic field. 

## Data Availability

Data are contained within the article or Appendix A.

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
