# Peer review of "Dy4, Dy5, and Ho2 Complexes of an N3O2 Aminophenol Donor: A Dy53-Peroxide Single Molecule Magnet"

_ijms, 2023, doi:10.3390/ijms24109061_

Round 1

Reviewer 1 Report

The present manuscript describes the synthesis, structural and spectral characterization of some new polynuclear complexes of Dysprosium(III) and Holmium(III) with pentadentate bis-aminophenolate ligand. The formation of different polynuclear patterns caused by the use of different starting metal salts (chlorides, nitrates) as well as by the different atmosphere conditions (on or without air oxygen) was shown. The authors have also provide complex investigations of the magnetic behaviour of new compounds. The unusual bis-peroxo-bridged pentanuclear compound [Dy5(H2L)2(H2.5L)2(NO3)43-O2)2]·2H2O (2·2H2O) has been shown to demonstrate SMM properties (Ueff barrier is 26 69.7 K). 

The manuscript is interesting and well-writted. I may advise its acceptance for publication after answers to some questioins and some minor revisions. 

The question: did the authors try to use holmium (III) chloride in the
same reaction instead of holmium nitrate?

More comments:

- Figure 5. Wrong caption, here compound 2 should be instead of compound 1.

- It seems (based on the given IR data) that compound 5 which was obtained from 4 on air is just a compex 2, possessing the same IR spectrum. Why do the authors assign the number 5 to this product but not 2?

-  p. 11, line 385. 1H NMR instead of 1H RMN.

- pleas pay attention to the carbon content in elemental analysis of complexes as well as ligand. In all cases, the C content was overestimated (from 0.2 to 0.7 abs%). I don't have doubts about the composition of compounds. However it seems that it is just a systematic mistake, and the CHN analyzer should be checked.

Author Response

The answers to the comments can be found in the attached document

Reviewer 2 Report

Vasquez et al. reported the synthesis and magnetic characterization of three polynuclear lanthanide complexes. One of these (number 2) exhibits ac in zero applied field.

I must say that I am myself very interested in lanthanide-based chemistry and magnetism and I therefore enjoy reading papers on the topic. I particularly appreciated table 1 and the related discussion. However, this paper seems a bit “unpolished” and it needs substantial rewriting to be published in a peer review journal. The chemistry is nice, but lacks of proper explanation. The analysis of the magnetic results is poor to say the least. The work that must be done is substantial, therefore my suggestion to the editor is to reject the paper in its current form and perhaps invite the authors to resubmit an entirely new version of the manuscript.

Hereafter some comments to back up my decision:

1)      At page 3 the authors state that only with Dy(NO3)3 (not with Ho(NO3)3) they can obtain the peroxo species. This is interesting but lacks a chemical explanation. This is an information that the reader wants to learn from the paper. Even more so since the authors attribute the formation of the peroxo to oxygen from the air (that is clearly present in both syntheses).

2)      This comment is related to the IR and Raman analysis of the complexes. If I am not mistaking, with two peroxos related by a centre of symmetry, group theory should give two modes, one in-phase which is Raman-active, and one out-of-phase which is (weakly) IR-allowed. Therefore I don’t see the argument in not trying Raman, even more so if the (weak) IR peak is observed. However, since I am not an expert here, please double check.

3)      The dc data are poorly discussed and sometimes the authors write down incorrect sentences. For example, I surely agree that the decrease in chit can be attributed to depopulation of the Stark levels and/or to AFM interactions, but in Ln clusters also FM interactions (typically of the order of less than 1 cm-1) can be present. The dominance of the crystal field over these interactions does not allow, in most cases, to judge “by eye” if they are present or not. Please remove the word AFM.

4)      The authors compare the saturation values of the magnetization to the ones expected for an axial species with the z axis oriented along the field. This is nonsense because they have measured a powder, not a single crystal. Of course the values that they get are different! An axial Dy3+ (or Ho3+, they have the same value) in a powder measurement does not give 10mub, but rather 5 mub due to the powder averaging. This also explains why the only derivative showing zero field relaxation is 2….the expected value is 4*5=20 mub and the authors obtain a similar value (26) while in the other two cases these values are very different (expected 2*5=10, 5*5=25, obtained 20 and 11). This suggests that the ground state is probably heavily mixed. This impacts the magnetic properties (see comment 7).

5)      The authors state that the alpha values that they get suggest multiple relaxation times (“more than one relaxation process at low temperature”). This is incorrect. The alpha values that they get are perfectly consistent with QTM at low T. Unsurprisingly, few lines later they state that the chi’’ vs T dependence “agrees with fast relax via quantum tunnelling”. This is correct.

6)      Where are the measurements of chi’’ of 1 below 6K? The QT should give a peak at ca. 3000 Hz, very stable between 2 and 6K. This peak is in the range of their experimental setup. Also, the authors stop their measurements at 13K. Why? The peak is still in the range. I suggest going a bit higher in T (at least arrive at the T where the peak disappears from the experimental window).

7)      Equation 1 contains the Raman term that is not used, please remove it. Also, plotting 1/tau vs T is useless in this case. The correct plot to show is lntau vs 1/T to correctly visualize the linear dependence (Orbach) and the plateau (QT). My impression is that the peak in chi’’ does not move that much, therefore the Orbach fit does not deliver a meaningful result. The aforementioned plot (and the further measurements that I suggest in comment 5) could help in determining if I am wrong.  

8)      The authors tried to measure the slow relaxation of all complexes in an applied field. This is of course meaningful. However they seem somehow “surprised” that an external field is unable to promote slow relaxation, i.e. they assume that their compounds are axial. They are not (see my comment number 3 on the saturation values of M). Of course, ab initio calculations or single crystal techniques could help finding the correct ground state composition.

ok.

Author Response

(The authors gave the same response as above.)
